# Caregivers of children with HIV in Botswana prefer monthly IV Broadly Neutralizing Antibodies (bNAbs) to daily oral ART

**Maureen Sakoi-Mosetlhi**[ID][1]*, **Gbolahan Ajibola**[ID][1], **Roxanna Haghighat**[2],
**Oganne Batlang**[1], **Kenneth Maswabi**[1], **Molly Pretorius-Holme**[3], **Kathleen M. Powis**[1,2,3,4],
**Shahin Lockman**[1,3,5], **Joseph Makhema**[1], **Mathias Litcherfeld**[5], **Daniel R. Kuritzkes**[6],
**Roger Shapiro**[1,3]

1 Botswana Harvard AIDS Institute Partnership, Gaborone, Botswana, 2 Harvard Medical School, Boston, Massachusetts, United States of America, 3 Department of Immunology and Infectious Diseases, Harvard T. H. Chan School of Public Health, Boston, Massachusetts, United States of America, 4 Departments of Internal Medicine and Pediatrics, Massachusetts General Hospital, Boston, Massachusetts, United States of America, 5 Ragon Institute of Mass General, MIT, and Harvard, Cambridge, Massachusetts, United States of America, 6 Brigham and Women's Hospital, Boston, Massachusetts, United States of America

* msakoi@bhp.org.bw

## Abstract

### Introduction

Monthly intravenous infusion of broadly neutralizing monoclonal antibodies may be an attractive alternative to daily oral antiretroviral treatment for children living with HIV. However, acceptability among caregivers remains unknown.

### Methods

We evaluated monthly infusion of dual bNAbs (VRCO1LS and 10–1074) as a treatment alternative to ART among children participating in the Tatelo Study in Botswana. Eligible children aged 2–5 years received 8–32 weeks of bNAbs overlapping with ART, and up to 24 weeks of bNAbs alone as monthly intravenous infusion. Using closed-ended questionnaires, we evaluated caregiver acceptability of each treatment strategy prior to the first bNAb administration visit (pre-intervention) and after the completion of the final bNAb administration visit (post-intervention).

### Results

Twenty-five children completed the intervention phase of the study, and acceptability data were available from 24 caregivers at both time points. Responses were provided by the child's mother at both visits (60%), an extended family member at both visits (28%), or a combination of mother and an extended family member (12%). Caregiver acceptance of monthly bNAb infusions was extremely high both pre-and post-intervention, with 21/24 (87.5%) preferring bNAbs to ART pre-intervention, and 21/25 (84%) preferring bNAbs post-intervention. While no caregiver preferred ART pre-intervention, 2/25 preferred it post-intervention. Pre-intervention, 3 (13%) caregivers had no preference between monthly bNAbs or

**Data Availability Statement:** All relevant data are within the manuscript and its Supporting information files. De-identified data used for this

analysis has been made available as supporting documents with this submission.

**Funding:** The author(s) received funding from National Institute of Allergy and Infectious Diseases, (U01AI135940), Dr Roger Shapiro.

**Competing interests:** The authors have declared that no competing interest exist.

**Abbreviations:** AIDS, Acquired Immune Deficiency Syndrome; ART, Antiretroviral treatment; bNAbs, Broadly neutralizing monoclonal antibodies; DAIDS, Division of AIDS; EIT, Early Infant Treatment; HIV, Human Immunodeficiency Virus; HRDC, Health Research and Development Committee; IV, Intravenous; PK, Pharmacokinetics; RNA, Ribonucleic Acid.

daily ART, and 2 (8%) had no preference post-intervention. Pre-intervention, the most common reasons for preferring bNAbs over ART were the perception that bNAbs were better at suppressing the virus than ART (n = 10) and the fact that infusions were dosed once monthly compared to daily ART (n = 9). Post-intervention, no dominant reason for preferring bNAbs over ART emerged from caregivers.

## Conclusions

Monthly intravenous bNAb infusions were highly acceptable to caregivers of children with HIV in Botswana and preferred over standard ART by the majority of caregivers.

## Clinical Trial Number

NCT03707977.

## Introduction

Pediatric HIV remains an important public health issue in resource-limited settings [1, 2]. Approximately 3.3 million children below the age of 15 are estimated to be living with HIV worldwide, with 90% in sub-Saharan Africa [3, 4]. Combination antiretroviral therapy (ART) is the mainstay of treatment in children living with HIV [5]. ART reduces plasma viremia, increases the CD4 cell count, and ultimately improves the health of children living with HIV. Despite the benefits and availability of ART, medication adherence remains suboptimal among children living with HIV due to multiple barriers, notably reliance on caregivers and guardians to regularly administer medication [6–8]. In addition, ART may be associated with toxicities and adverse side effects that include anaemia [9], metabolic abnormalities [10–12], and reduced growth [10, 11]. Long-acting, less toxic strategies for managing pediatric HIV may offer an important advance over current strategies.

Broadly neutralizing monoclonal antibodies (bNAbs) are a potential therapeutic option for HIV when used in combination [13–15]. bNAbs target specific viral epitopes and have the potential to activate the host's immune responses [13, 16, 17] and neutralize diverse circulating variants of HIV-1 generated in rare subsets of persons living with HIV [18]. Passive administration of bNAbs in individuals living with HIV has been effective in suppressing viremia [18, 19]. In the past 5 years, several bNAbs have been evaluated in clinical trials for HIV treatment including: VRC01, VRC01-LS, 3BNC117, 3BNC117-LS, 10–1074 and 10-1074-LS [13, 16, 20]. Most of these clinical trials have focused on establishing the safety and pharmacokinetic profiles of bNAbs, and all bNAbs tested thus far have been found to be safe and tolerable, with optimal doses established for most [13, 21–25].

However, the vast majority of bNAb studies have been conducted among adults living with HIV, with very few studies in children [26–28]. Furthermore, no study, to date, has evaluated caregiver acceptability of bNAbs as a treatment option for children living with HIV. We assessed the acceptability of monthly dual bNAb infusions among caregivers of early-treated children living with HIV enrolled in the Botswana-based Tatelo Study, a dual bNAb treatment study [28]. Findings from our study provide insight into caregivers' perspectives on this novel form of treatment, identify salient concerns for caregivers that may influence the uptake of bNAbs for children, and highlight the priorities and unmet needs of caregivers of children receiving bNAbs.

## Materials and methods

### Study design

The Tatelo study was a phase I/II, multi-site clinical trial of treatment with VRC01LS and 10–1074, offered to children living with HIV-1 and virally suppressed. Following an initial pharmacokinetic (PK) step, the main study occurred in three steps (steps 1–3). In Step 1, ART was continued with monthly VRC01LS and 10–1074 treatment. In Step 2, ART was stopped and participants remained on monthly bNAbs for the maintenance of viral suppression for up to 24 weeks. While on dual bNAb-only treatment, virologic monitoring occurred weekly for the first month, then every two weeks from 6–24 weeks. Participants who experienced viral rebound ≥400 copies/mL in step 2 had bNAbs stopped and ART restarted immediately. In step 3, dual bNAbs were discontinued and participants were re-started on ART and followed for 24 weeks.

VRC01LS and 10–1074 were administered as monthly intravenous infusions. Infusions were administered over 1 hour, with initial infusion of 10–1074 followed by VRC01LS at each visit. bNAbs were prepared using aseptic procedures by trained and certified pharmacists, and administration was conducted by a trained nurse or physician.

### Study setting

The Tatelo study was conducted at two Division of AIDS (DAIDS) registered clinical research sites in Botswana (on the premises of Princess Marina Hospital, Gaborone, and Nyangabgwe Hospital, Francistown). Study recruitment and intervention occurred between 13 June 2019 and 03 December 2021.

### Participants

Acceptability of dual bNAb infusions was assessed among primary caregivers of children who completed at least Step 1 of the Tatelo study. Children who participated in the Tatelo study were selected from a cohort of early-treated children living with HIV, on ART for ≥96 weeks at enrollment, and documented to have viral suppression (HIV RNA <40 copies/mL) at least 24 weeks prior to study enrollment. Further inclusion criteria for the Tatelo Study included the ability to remain within the study catchment area for close follow-up for at least 12 weeks, caregiver willingness for the child to receive IV infusions of bNAbs, and caregiver willingness to provide signed informed consent. All Tatelo participants had previously taken part in the Early Infant Treatment [29] study, a clinical trial evaluating early infant HIV-1 diagnosis and treatment in Gaborone and Francistown, Botswana (NCT02369406) [27].

### Data collection and definitions

Primary caregivers answered closed-ended questionnaires prior to the first bNAb administration (pre-intervention) and after the completion of the final bNAb administration or at the time of viral rebound for those with HIV RNA >400 copies/mL while on bNAb only (post-intervention). Primary caregivers were classified as "immediate family" (participant's mother or father), "extended family" (participant's grandparent, older sibling, aunt, or uncle), or "both" (if respondents were both immediate and extended family). Caregiver acceptability of intervention was assessed by asking for treatment preference if medical benefits were equal, at both pre- and post-intervention time points. Caregivers were also probed for factors that would make them more likely to prefer monthly bNAb infusions over daily ART. Intervention success was defined as maintenance of HIV RNA <400 copies/ml for 24 weeks while on dual bNAb-only treatment.

## Data analysis

Caregivers' responses to close-ended questionnaire items were described using summary statistics. Free-text responses to questionnaire items were coded and summarized using descriptive thematic analysis. Coded themes were analyzed statistically using SPSS version 27.

## Ethical considerations

Ethical approval was obtained from institutional review boards locally in Botswana from the Ministry of Health's Health Research and Development Committee (HRDC) and in the US by the Harvard T.H. Chan School of Public Health IRB. Study participation was voluntary, and written informed consent for participation was obtained from primary caregivers.

## Results

Twenty-five children participated in all three study steps and were included in this analysis. Of these, 11(44%) were enrolled in the Gaborone site while 14 (56%) were enrolled at the Francistown site. The median age at enrollment into Step 1 was 3.6 years, with a range of 2.4 to 5.6 years; 16 (64%) were females. Twelve (48%) of the children were brought to the clinic by immediate family members only, 5 (20%) by extended family members only, and 8 (32%) by both immediate and extended family members. Maternal and child characteristics at entry into step 1 are shown in (Table 1).

In total, 14 (56%) of the 25 enrolled children had viral rebound on dual bNAb, while 11 (44%) maintained viral suppression for 24 weeks. Acceptability data were available from 24 caregivers pre- and post-intervention (1 primary caregiver was unavailable at the pre-

**Table 1. Entry characteristics among Tatelo participants.**

| Child Characteristics | | |
|---|---|---|
| **Enrollment site; N (%)** | Gaborone | 11 (44.0) |
| | Francistown | 14 (56.0) |
| **Age in years; median (range)** | | 3.6 (2.4, 5.6) |
| **Sex; N (%)** | Female | 16 (64.0) |
| | Male | 9 (36.0) |
| **Virologic failure while on dual bNAb treatment only; N (%)** | Yes | 14 (56.0) |
| | No | 11 (44.0) |
| **Primary caregiver; N (%)** | Immediate family members only | 12 (48.0) |
| | Extended family members only | 5 (20.0) |
| | Both | 8 (32.0) |
| **Maternal Characteristics*** | | |
| **Maternal age in years; median (range)** | | 29 (25, 32) |
| **Marital Status; N (%)** | Never married | 22 (95.7) |
| | Married | 1 (4.3) |
| **Highest level of education; N (%)** | None | 1 (4.3) |
| | Primary | 5 (21.7) |
| | Secondary | 13 (56.5) |
| | Tertiary | 4 (17.4) |
| **Occupation; N (%)** | Employed | 5 (21.7) |
| | Unemployed | 18 (78.3) |
| **Earnings; N (%)** | None | 6 (26.1) |
| | Some earning | 17 (73.9) |

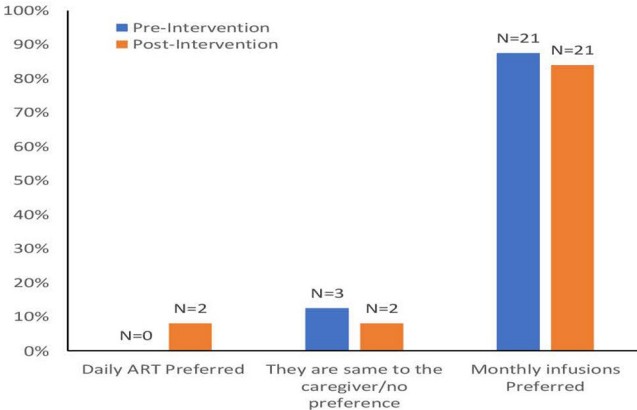

**Fig 1. Pre- and post-intervention preferences by caregivers for monthly bNAb infusions compared to daily ART (if the medical benefit were equal).**

intervention visit). Responses were provided by the child's mother at both visits (60%), an extended family member at both visits (28%), or a combination of mother and extended family member (12%). Caregiver acceptance of bNAbs was high both pre-and post-intervention, with 21/24 (87.5%) preferring bNAbs to ART pre-intervention, and 21/25 (84%) preferring bNAbs post-intervention (9 cared for a child who remained virally suppressed, 12 for a child with a viral rebound on bNAbs). While no caregiver preferred ART pre-intervention, 2/25 preferred it post-intervention (1 cared for a child who remained virally suppressed, 1 for a child with viral rebound on bNAbs). Pre-intervention, 3 (13%) caregivers had no preference between bNAbs or ART, and 2 (8%) had no preference post-intervention. (Fig 1) illustrates pre-and post-intervention preferences among caregivers for monthly bNAb infusions compared to daily ART when asked to assume that the medical benefit of either treatment option was equal. All participants who experienced a viral rebound in the intervention phase of the study were immediately re-initiated on ART and all re-suppressed to levels <40 copies/mL at a median of 4 weeks (range: 1 to 20 weeks).

Factors that would lead caregivers to prefer monthly bNAb infusions over daily ART were also explored (Table 2). Pre-intervention, the most commonly cited factors to favor bNAb were efficacy in suppressing HIV (41.7%, n = 10) and monthly rather than daily dosing (37.5%, n = 9). Post-intervention, the most frequently reported reason to favor bNAb was monthly dosing (37.5%, n = 9), although treatment efficacy (20.8%, n = 5) and single injection infusions (20.3%, n = 5) were also cited.

**Table 2. Descriptive themes of caregiver responses pre- and post-intervention.**

| Reasons for preferring monthly infusions over daily ART; N (%) | Pre–bNAbs | Post–bNAbs |
|---|---|---|
| If infusions were better at suppressing the virus than ART | 10 (42%) | 5 (21%) |
| If infusions continued to be once monthly compared to daily ART | 9 (38%) | 9 (38%) |
| If infusions were every 3 months instead of every month | 3 (13%) | 2 (8%) |
| If infusion visits were shorter | 1 (4%) | 2 (8%) |
| If the clinic were closer | 1 (4%) | - |
| If infusions were a single injection | - | 5 (20%) |
| Nothing, will always prefer daily ART | - | 1 (4%) |

## Discussion

Monthly intravenous infusion of bNAbs may be an attractive and feasible alternative to daily ART for children living with HIV, given the numerous barriers to treatment adherence with daily ART administration. In this first pediatric treatment study to use monthly intravenous infusion of bNAbs instead of standard ART, we found that bNAbs were highly acceptable to caregivers. Our findings suggest that caregivers would prefer monthly infusions over daily ART if treatments were equally effective. This preference was consistent both before and after the intervention of bNAb-only treatment. Our findings support reports from the World Health Organization [30] that highlight the potential advantages of bNAbs over daily ART. These advantages may have played a role in caregiver preferences.

In our study, the caregivers' preferences for monthly bNAb infusions over daily ART seemed to be influenced by several factors, but no dominant factor emerged at the end of the trial. Reasons cited by caregivers included less frequent dosing schedule (monthly) as compared to daily ART, and an interest in using a treatment that might be better at suppressing the virus than standard ART. In this cohort, as in many pediatric cohorts, adherence to daily ART was challenging, with over 50% of the children experiencing viral rebound related to poor adherence at some point in the first 2 years of life [31]. These findings emphasize the desire by caregivers to have a more flexible, less intensive, and at least equally effective form of treatment compared with ART that is less likely to affect their day-to-day routine and improve adherence. Our data suggest that caregivers are willing to give up several hours once a month to achieve this goal if it eliminates the burden of daily ART.

Our study has some limitations. The generalizability of our findings may be hindered by the small number of children enrolled, and while we believe our cohort is highly representative of children living with HIV and their caregivers in Botswana, it may not reflect values and preferences elsewhere. For convenience and because of time restrictions, we used a closed-ended questionnaire with the ability for some open-ended responses. However, we recognize that this may be a less sensitive tool than an entirely qualitative assessment of caregiver preferences.

## Conclusions

Children largely rely upon caregivers and guardians to provide medication until adolescence. Additionally, lifetime daily ART increases the risk for medication-related toxicities, which can discourage treatment adherence. As more studies explore the efficacy and feasibility of bNAbs for pediatric HIV treatment, it is critical to understand the acceptability of this form of treatment among caregivers. We demonstrated that monthly bNAb infusions were highly acceptable among caregivers of children living with HIV in Botswana, with most caregivers preferring this form of treatment over daily ART if treatments were equally effective. Our findings suggest that caregiver acceptability is unlikely to be a barrier to bNAb uptake and eventual programmatic use for children living with HIV. As bNAb treatment for HIV expands, further research in a larger and more diverse population is required to fully understand other barriers and facilitators to the uptake of this novel treatment by caregivers of children living with HIV.

## Supporting information

**S1 Data.**
(XLSX)

**S2 Data.**
(PDF)

**S1 Checklist.**
(DOCX)

## Acknowledgments

We would like to thank the Tatelo Study participants and their families. We thank the Tatelo and EIT Study teams and collaborators, including Dorcus Babuile, Rachel Bowman, Caroline Brackett, Loveness Bunhu, Alex Carnacchi, Lars Colson, Trevor Cordwell, Jack Disaro, Lorato Esele, Tshepho Frank, Olebile Kgakge, Tsholofelo Kebopetswe, Nametso Kelentse, Bob Lin, Judith Lucas, Kaia Lyons, Abraham Maigwa, Ria Madison, Princess Mapenshi, Mogomotsi Matshaba, Simon Masopa, Sam McMillan, Charlotte Mdluli, Lendsey Melton, Mompati Mmalane, Maureen Mosetlhi, Akanyang Motlhanka, Sarah Mudrak, Sandeep Narpala, Muhammed Naqvi, Thabani Ncube, Sandra Ndongwe, Martha Ngwaca, Maduo Oabona, Salome Othusitse, Gaoele Pelontle, Obonwe Pule, Lynette Purdue, Christina Reding, Marcella Sarzotti-Kelsoe, Tumalano Sekoto, Ngozana Seonyatseng, Dineo Tumagole, and Joshua Weiner; and, at Labcorp-Monogram Biosciences, Christos Petropoulos, Kristi Strommen, Yolanda Lie, Tim Persyn, and the Clinical Reference Lab. We also thank the Botswana Ministry of Health and Wellness and the Tatelo Safety Monitoring Committee, including Grace John-Stewart, Katherine Luzuriaga, Loeto Mazhani, Pablo Tebas, Terry Fenton, and Jane Lindsey.

**Disclaimer:** The contents of this article are solely the responsibility of the authors and do not necessarily represent the official positions of the funding agencies.

## Author Contributions

**Conceptualization:** Maureen Sakoi-Mosetlhi, Gbolahan Ajibola, Kathleen M. Powis, Roger Shapiro.

**Data curation:** Maureen Sakoi-Mosetlhi, Gbolahan Ajibola, Oganne Batlang, Kenneth Maswabi.

**Formal analysis:** Maureen Sakoi-Mosetlhi, Gbolahan Ajibola, Roxanna Haghighat.

**Funding acquisition:** Roger Shapiro.

**Supervision:** Roger Shapiro.

**Writing – original draft:** Maureen Sakoi-Mosetlhi.

**Writing – review & editing:** Gbolahan Ajibola, Roxanna Haghighat, Oganne Batlang, Kenneth Maswabi, Molly Pretorius-Holme, Kathleen M. Powis, Shahin Lockman, Joseph Makhema, Mathias Litcherfeld, Daniel R. Kuritzkes, Roger Shapiro.

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
