## [Decision Letter · Decision Letter 0]

19 Dec 2023

PONE-D-23-35663Caregivers of Children with HIV in Botswana Prefer Monthly IV Broadly Neutralizing Antibodies (bNAbs) to Daily Oral ARTPLOS ONE

Dear Dr. SAKOI-MOSETLHI,

Thank you for submitting your manuscript to PLOS ONE. After careful consideration, we feel that it has merit but does not fully meet PLOS ONE’s publication criteria as it currently stands. Therefore, we invite you to submit a revised version of the manuscript that addresses the points raised during the review process.

We look forward to receiving your revised manuscript.

Kind regards,

Jayanta Bhattacharya

Academic Editor

PLOS ONE

Journal Requirements:

3. In the online submission form, you indicated that Deidentified data used for this analysis will be shared with interested researchers upon request.

Data not publicly available as the study is still ongoing. 

4. Please amend your manuscript to include your abstract after the title page.

Reviewers' comments:

Reviewer's Responses to Questions

**Comments to the Author**

1. Is the manuscript technically sound, and do the data support the conclusions?

Reviewer #1: Partly

Reviewer #2: Partly

2. Has the statistical analysis been performed appropriately and rigorously? 

Reviewer #1: Yes

Reviewer #2: Yes

3. Have the authors made all data underlying the findings in their manuscript fully available?

Reviewer #1: No

Reviewer #2: Yes

4. Is the manuscript presented in an intelligible fashion and written in standard English?

Reviewer #1: Yes

Reviewer #2: Yes

5. Review Comments to the Author

Reviewer #1: This is a small study focussing on assessing acceptability of once a month bNAbs as against daily antiretroviral therapy among caregivers of 25 HIV infected children. These children were those who were put on ART very early in life and were virologically suppressed. They also had participated in some of the studies under taken by the investigators. bNAbs are likely to become an important area of work in management of HIV disease. Children depend on their parents, more often mothers or other caregivers to provide antiretroviral therapy everyday. Low adherence to ART is common among children. This study finds that monthly bNAb regimen was more acceptable to caregivers compared to daily ART regimen. The authors mention a limitation that it had more close ended questions. However, there are certain areas which require clarifications;

1. All the children and thus their caregivers included in the study had been participating one or the other study protocols, it reduces generalisability of findings additionally.

2. The responses could have been different if the caregivers were informed about the plasma viral loads on real time basis/ or at least before the post-assessment.

3. A significant proportion of children had to be shifted from bNAbs to antiretrovirals again, it is unclear whether the caregivers were informed of the reasons. It would have helped if their responses would have been recorded then as well. It helps in understanding whether care givers convenience or their concern about viral suppression influenced their post-assessment response. This is partly reflected in responses to a question “if infusions were better at suppressing the virus than ART”.

Reviewer #2: In this study, Ajibola G et al have evaluated monthly infusion of dual bNAbs (VRCO1LS and 10-1074) as a treatment alternative to ART among children living with HIV-1, who are virally suppressed by ART (HIV RNA 68 <40 copies/mL) at least 24 weeks prior to enrolment in the Tatelo Study in Botswana. Primary caregivers of 25 children enrolled in the study answered closed-ended questionnaires prior to the first bNAb administration (pre-intervention) and after the completion of the final bNAb administration (post-intervention). 14 (56%) of the 25 enrolled children had viral rebound on dual bNAb, while 11 (44%) maintained viral suppression for 24 weeks. Acceptability data were available from 24 out of the 25 caregivers pre- and post-intervention. Pre-intervention, the most common reasons for preferring bNAbs over ART were the perception that bNAbs were better at suppressing the virus than ART (n=10) and the fact that infusions were dosed once monthly compared to daily ART (n=9). Post-intervention, no dominant reason for preferring bNAbs over ART emerged from caregivers. 2/25 preferred ART post-intervention (1 for a child who remained virally suppressed, 1 for a child with viral rebound on bNAbs). Intervention success was defined as maintenance of HIV RNA <400 copies/ml for 24 weeks while on dual bNAb-only treatment. The authors report that monthly bNAb infusions were highly acceptable among caregivers of children living with HIV in Botswana, with most caregivers preferring this form of treatment over daily ART if treatments were equally effective and suggest that caregiver acceptability is unlikely to be a barrier to bNAb uptake and eventual use of bnAbs to treat children living with HIV.

The study findings are supportive towards the acceptability of caregivers of the bnAb based therapy in children living with HIV in Botswana. However there are few concerns, some of which have been partly addressed by the authors and should be detailed as limitations in the present study.

1. Being a close ended questionnaire based study, the extent of understanding of the care givers, about the implications of viral rebound in 14 children who were on dual bNAb, is not addressed. Intervention success was defined as maintenance of HIV RNA <400 copies/ml for 24 weeks while on dual bNAb-only treatment. The viremic status of children who had progressed to viral copies >40 from the time of recruitment would be clear only on further follow up beyond the course of this study.

Whether these caregivers were counselled and informed of the above implications at the pre and post intervention stages prior to answering the questionnaire is not clear.

2. The education status and socioeconomic conditions of the caregivers, and the inclusion of subjects only within the study catchment area for close follow-up for at least 12 weeks, may be confounding factors that limit the observations and need to be addressed in detail as limitations of the study.

3. The sample size is vey small and the observations should be described as preliminary findings that need to be validated in a larger cohort.

6. PLOS authors have the option to publish the peer review history of their article (what does this mean?). If published, this will include your full peer review and any attached files.

Reviewer #1: No

Reviewer #2: No

---

## [Author Response · Author response to Decision Letter 0]

15 Feb 2024

POINT-BY-POINT RESPONSE TO REVIEWER COMMENTS

Reviewer #1

1. All the children and thus their caregivers included in the study had been participating one or the other study protocols, it reduces the generalisability of findings additionally.

Response: We appreciate the reviewer’s concerns and acknowledge that working with established cohorts from previous studies within a research setting could be a potential limitation as it provides less information on the real-life applicability of our intervention. However, as with all caregivers caring for children living with HIV, our participants face similar challenges with daily ART administration and have similar desires for simplified/alternative treatment approaches as we have emphasized in the manuscript (Line 33-39). In addition, any child receiving bNAbs in the foreseeable future will be a similar study setting, and studies such as Tatelo will be the only source of acceptability information to help determine the feasibility of this strategy. Therefore the findings of this study can be useful and applicable to other caregivers who find themselves caring for children living with HIV in Botswana and beyond.

2. The responses could have been different if the caregivers were informed about the plasma viral loads on real real-time basis/ or at least before the post-assessment.

Response: We thank the reviewer for this comment and confirm that all caregivers were informed of their child’s plasma viral load (near real-time) at every scheduled visit per the study procedures. For those who experienced viral rebound during the bNAb only treatment phase, caregivers were contacted immediately and invited to the study clinic where they were notified of the occurrence of the rebound and the implication to their child’s inability to continue in the bNAb-only phase of the study. Upon this notification, study phase exit visits were conducted including administration of the post-intervention acceptability questionnaire. We have modified the data collection and definitions section of the manuscript to reflect these details in lines 99-102 which now reads “Primary caregivers answered closed-ended questionnaires prior to the first bNAb administration (pre-intervention) and after the completion of the final bNAb administration or at the time of viral rebound for those with HIV RNA >400 copies/mL while on bNAb only (post-intervention). 

3. A significant proportion of children had to be shifted from bNAbs to antiretrovirals again, it is unclear whether the caregivers were informed of the reasons. It would have helped if their responses would have been recorded then as well. It helps in understanding whether caregivers convenience or their concern about viral suppression influenced their post-assessment response. This is partly reflected in responses to a question “if infusions were better at suppressing the virus than ART”.

Response: All children who rebounded while on the intervention phase of the study (monthly dual bNAbs only) were switched back to ART and monthly dual bNAbs were permanently discontinued. Information on caregiver acceptability was collected and assessed at the end of the intervention phase for those with no viral rebound or at the time of viral rebound for those with HIV RNA >400 copies/mL while on dual bNAb only. As highlighted in 2 above, we have updated the study methods to further clarify that caregivers were informed of their child’s HIV plasma RNA levels at every study visit as well as the reason for the permanent discontinuation of bNAbs where appropriate. 

Reviewer #2

1. Being a close-ended questionnaire-based study, the extent of understanding of the caregivers, about the implications of viral rebound in 14 children who were on dual bNAb, is not addressed. Intervention success was defined as maintenance of HIV RNA <400 copies/ml for 24 weeks while on dual bNAb-only treatment. The viremic status of children who had progressed to viral copies >40 from the time of recruitment would be clear only on further follow-up beyond the course of this study. Whether these caregivers were counselled and informed of the above implications at the pre and post-intervention stages prior to answering the questionnaire is not clear.

Response: Caregivers were made aware of the possibility of their children rebounding while on dual bNAb-only treatment, and its implications for permanent discontinuation of the bNAbs and re-introduction of daily ART. Caregivers of participants who rebounded were given this information as part of the informed consent process and at the time of viral rebound. All the participants who were taken off the intervention step due to detectable plasma HIV RNA were followed weekly until they attained viral suppression and for up to 24 weeks post bNAbs cessation. We have now reported on the time to re-suppression for all participants who were re-initiated on ART before completing the dual bNAb-only treatment phase due to HIV RNA >400 copies/mL. Lines 147 – 149 of the manuscripts now read “All participants who experienced a viral rebound in the intervention phase of the study were immediately re-initiated on ART and all re-suppressed to levels <40 copies/mL at a median of 4 weeks (range: 1 to 20 weeks)”. 

2. The education status and socioeconomic conditions of the caregivers, and the inclusion of subjects only within the study catchment area for close follow-up for at least 12 weeks, may be confounding factors that limit the observations and need to be addressed in detail as limitations of the study.

Response: We thank the reviewer for this comment and agree that this could potentially limit the generalizability of our findings. However, Table 1 of the manuscript describes the basic socio-demographics of study participants and shows that participants were enrolled from different geographical locations and socioeconomic backgrounds, reducing the possibility of bias. Because the cohort was created by screening nearly all HIV exposed children in the catchment areas, and these areas covered ~ 30% of all exposed children in Botswana during the initial accrual period, and almost 100% of the children who screened positive enrolled, the demographics of the Tatelo cohort are highly representative of children living with HIV in Botswana.

3. The sample size is very small and the observations should be described as preliminary findings that need to be validated in a larger cohort.

Response: We agree with the reviewer on the need for the validation of our findings in other cohorts of early-treated infants globally and have included this in our conclusion. Lines 206 - 208 of our conclusion state the need for further research in a larger and more diverse population as encouraged by the reviewer.

---

## [Editor Report · Decision Letter 1]

20 Feb 2024

Caregivers of Children with HIV in Botswana Prefer Monthly IV Broadly Neutralizing Antibodies (bNAbs) to Daily Oral ART

PONE-D-23-35663R1

Dear Dr. SAKOI-MOSETLHI,

We’re pleased to inform you that your manuscript has been judged scientifically suitable for publication and will be formally accepted for publication once it meets all outstanding technical requirements.

Kind regards,

Jayanta Bhattacharya

Academic Editor

PLOS ONE
---

## [Editor Report · Acceptance letter]

16 Mar 2024

PONE-D-23-35663R1 

PLOS ONE

Dear Dr. Sakoi-Mosetlhi, 

I'm pleased to inform you that your manuscript has been deemed suitable for publication in PLOS ONE. Congratulations! Your manuscript is now being handed over to our production team.

Kind regards, 

on behalf of

Dr. Jayanta Bhattacharya 

Academic Editor

PLOS ONE